# Impact of Artificial Intelligence on Spectator Viewing Behavior in Sports Events: Mediating Role of Viewing Motivation and Moderating Role of Player Identification

**DOI:** 10.3390/bs15121702

**Published:** 2025-12-08

**Authors:** Jie Min, Qing Xie, Yongjian Liu

**Affiliations:** 1School of Management, Wuhan University of Technology, Wuhan 430070, China; mjack@zuel.edu.cn; 2Sports Department, Zhongnan University of Economics and Law, Wuhan 430073, China; 3School of Computer Science and Artificial Intelligence, Wuhan University of Technology, Wuhan 430070, China; felixxq@whut.edu.cn

**Keywords:** sports technology, AI-enabled experience, viewing motivation, player identification, moderated mediation

## Abstract

With the widespread application of artificial intelligence (AI) technology in the sports industry, the spectator’s experience is increasingly shaped by AI-driven features. To explore the mechanism through which the perceived AI-enabled spectating experience affects viewing behavior, and to validate the mediating role of viewing motivation (SDT Needs Satisfaction) in the relationship between AI and viewing behavior as well as the moderating role of player identification in this mediating pathway, we adopted literature review, survey, and empirical analysis methods. A sample of 272 Chinese tennis enthusiasts was surveyed, and both the measurement model and the structural model were evaluated. The results indicate that the measurement model has good internal consistency, reliability, convergent validity, and discriminant validity. The perceived AI-enabled spectating experience has a significant positive effect on viewing motivation, viewing intention, and recommendation intention. The data show that the indirect effect of the perceived AI-enabled spectating experience on the viewing intention through the viewing motivation is 0.0479, and the indirect effect of the perceived AI-enabled spectating experience on the recommendation intention through the viewing motivation is 0.0548. Both reached a significant level, and the direct effect of the perceived AI-enabled spectating experience has also reached statistical significance. Therefore, viewing motivation plays a partial mediating role between AI and viewing intention and between AI and recommendation intention. Player identification plays a significant positive moderating role (β = 0.2809 on viewing intention, β = 0.1621 on recommendation intention) in the relationship between viewing motivation and viewing behavior; however, it does not moderate the relationship between AI and viewing motivation. In other words, for spectators with higher player identification, viewing motivation drives more strongly both their viewing intention and recommendation intention. We suggest that sports event organizers and media use AI technologies to design differentiated marketing to enhance user engagement and optimize spectators’ experience. For spectators with lower player identification, improving service quality can enhance their satisfaction; for those with higher player identification, efforts should focus on strengthening their connection with the players.

## 1. Introduction

The rapid development of artificial intelligence has greatly affected the decision making behavior of individual consumers. The growing convergence of AI in sports industry marketing, including everything from personalized content recommendations to real-time data analytics and interactive fan engagement tools, represents a new revolution in the way companies engage customers, especially on social media. This shift makes the audience of the event no longer a passive consumer but an active participant whose preferences, motivations, and behaviors are increasingly influenced by AI-driven innovation.

The decision making behavior of the audience after the game, including recommending and watching the game again, is an important issue in the field of game consumption that has long been a concern of game researchers and organizers. At present, some scholars have confirmed the role of AI in enhancing user experience. For instance, Dr. Muhammad Asif proposed that by leveraging various artificial intelligence technologies, such as natural language processing, machine learning, and computer vision, user engagement, satisfaction, and other experiences could be enhanced. But it is still unclear how spectators’ perception of these AI-driven features affects their viewing motivation and subsequent behavior. In sports events, the audience is the main consumer group of the core products of the event and also the key factor to enhance the value of the derivatives of the event. While SDT has been applied to general technology adoption, the AI-enabled sports spectating context represents a unique scenario. Unlike static media, AI offers instantaneous cognitive enhancement (satisfying competence through real-time data) and dynamic interactive choices (satisfying autonomy through personalization). Our study is the first to systematically model the AI-enabled spectating experience as a coherent set of social–environmental factors driving SDT fulfillment, thereby bridging a critical gap between the technology use literature and classical SDT theory. Therefore, it is of great significance to study the relationship between the audience’s motivation and behavioral intention to enhance their experience of watching the event and their willingness to watch the event again.

Some studies conducted by scholars both domestically and internationally have confirmed that from real-time game analysis to personalized experiences supported by complex algorithms, artificial intelligence is completely transforming the way spectators interact with their favorite sports events ([15]). However, the underlying mechanisms of these effects remain unclear. In view of the importance of artificial intelligence for the audience’s motivation and its potential impact on the audience’s decision making behavior, it is particularly important to explore the mechanism by which the perceived AI-enabled spectating experience affects the viewing behavior of spectators. Considering the above factors, in this study, we aim to answer the following questions:

1: What is the impact of the perceived AI-enabled spectating experience on the audience’s motivation to watch the game, their willingness to watch the game, and their willingness to recommend?

2: What role does viewing motivation (SDT Needs Satisfaction) play in AI and the willingness to watch the game and the willingness to recommend?

3: Does player identification moderate the relationship between the perceived AI-enabled spectating experience and viewing motivation (SDT Needs Satisfaction) or the relationship between viewing motivation (SDT Needs Satisfaction) and viewing behavior?

Therefore, this paper aims to verify the causal path relationship between AI and watching behavior through empirical research and to enhance the generalization ability and explanatory power of existing theoretical models. In addition, this paper takes “Viewing Motivation (SDT Needs Satisfaction)” and “player identity” as mediating variables and moderating variables, further explores the impact of incentives, and then enriches the research perspective in the field of sports event consumption, laying the foundation for follow-up research.

## 2. Literature Review

### 2.1. Definition and Development of Artificial Intelligence

On the one hand, this dilemma stems from the fact that “intelligence” itself is a complex cognitive phenomenon, and its philosophy and psychology have not yet reached a consensus. On the other hand, it is because of the rapid development of the field of artificial intelligence. As a result, machine behavior, which was regarded as “intelligent” just five years ago, has rapidly become obsolete and almost meaningless ([21]). Faced with this dynamic definition dilemma, researchers usually define AI by establishing a mapping relationship with human intelligence.

Under this framework, human intelligence is described as “the bio-psychological potential to process information, solve problems, or create valuable products in a given culture” ([16]). In 1955, the founder of AI in the Dartmouth Research Project defined it as “the science of making machines operate in a way that would be considered intelligent if humans exhibited the same behavior” ([28]). Similarly, Marvin Minsky, a cognitive scientist and AI pioneer, defined AI as “the science of enabling machines to perform tasks that require human intelligence” ([29]).

Among many definitions, this paper adopts the version proposed by the European Union High-level Artificial Intelligence Expert Group ([19]), which has been adopted by the European Union AI Monitoring Agency AI Watch ([32]), as follows:

“Artificial intelligence system is a software system designed by human beings, which perceives the environment through data, parses structured or unstructured data, and decides the best action plan to achieve the set complex goals based on knowledge reasoning or information processing.”

The development process of AI shows significant cyclical fluctuation characteristics. According to the stage division of [12] ([12]), the evolution of AI can be deconstructed into three key periods.

#### 2.1.1. Enlightenment Period (1950s–1970s)

During this period, the core feature was the construction of an algorithmic foundation, and the research community was full of optimistic expectations for computer simulations of human thinking. In 1961, Rosenblatt proposed the perceptron model, which laid the theoretical foundation for connectionism and neural networks (NN). The ELIZA system developed by Weizenenbaum in 1966 was the first to demonstrate the feasibility of natural language processing, although the fragility of its dialogue logic eventually exposed the limitations of the technology. Driven by radical declarations from academia (such as Minsky’s prediction that general artificial intelligence will be realized within ten years), the government launched large-scale investments. However, due to the bottleneck of computing power and memory, the early AI system had trouble fulfilling its promise, coupled with the crisis of trust caused by excessive hype, which eventually led to the first AI winter in the 1970s ([12]).

#### 2.1.2. Knowledge Engineering Period (1970s–1990s)

In order to break through the constraints of computing power, the focus of artificial intelligence research has shifted to symbolic artificial intelligence. The expert system based on Lisp/Prolog language simulates human expertise through a rule base and reasoning engine and has achieved commercial success in medical diagnosis, industrial control, and other fields. However, the system gradually exposed a knowledge acquisition bottleneck (the cost of domain expert participation is too high) and the system rigidity defect (the rule update lags behind the display complexity) ([44]). Eventually, as maintenance costs rise and returns on investment fall, AI research falls into a low ebb again. However, it is worth noting that the introduction of interdisciplinary mathematical tools (such as Bayesian networks) at this stage foreshadows the subsequent development of machine learning.

#### 2.1.3. Data-Driven Period (1990s–Present)

The paradigm revolution began with the deep belief network proposed by Hinton’s team, and the real tipping point was AlexNet’s record 16% classification error rate in the ImageNet competition in 2017 ([22]). Deep learning aims to automatically extract feature representation through a multi-layer neural network, which solves the pain point of traditional machine learning relying on artificial feature engineering ([23]).

Technological breakthroughs have driven dramatic changes in capital flows, and private enterprises have replaced governments as the main investors in AI ([12]), such as Google’s hiring of Geoffrey Hinton and Facebook’s hiring of Yann LeCun. Microsoft invested USD 10 billion in OpenAI. Similarly, many Chinese companies have revealed their strategic plans in the field of artificial intelligence through major investment proposals. The global competitive situation is gradually emerging; the United States relies on the Silicon Valley innovation ecosystem to pursue the advantages of algorithms and hardware ([32]). The State Council of China has developed an ambitious three-stage plan to achieve the national AI goal outlined by [13] ([13]) to establish China as the “world’s major AI innovation center” by 2030. The EU leads in the volume of academic output, with member states working with the European Commission (EC) to allocate significant funding for AI investments ([12]).

### 2.2. Artificial Intelligence Classification

In the field of management research, [6] ([6]) and [25] ([25]) generally believe that the classification of artificial intelligence systems can draw lessons from the three-dimensional structure of human intelligence: cognitive intelligence, emotional intelligence, and social intelligence.

Before discussing AI systems, it is important to distinguish between true AI and expert systems. The latter can be viewed as a series of rules written by humans in the form of if–then statements ([21]), which do not strictly belong to AI because they cannot learn from external data themselves. At present, artificial intelligence can be divided into three different stages: narrow artificial intelligence system, artificial general intelligence, and super intelligence. Most of the AI systems we have today fall into the narrow AI (ANI) category, also known as “weak” AI. Although these systems surpass humans in specific tasks, their machine learning modes are still subject to preset task boundaries ([12]).

On the other hand, artificial general intelligence (AGI), also known as “strong” artificial intelligence, refers to the development of machines with human intelligence. The goal of AGI is to enable machines to perform any intellectual task that humans can perform, such as reasoning, learning, and problem solving ([12]). Although [21] ([21]) and [12] ([12]) have suggested that this technology may appear in the distant future, the breakthrough of GPT series models indicates the dawn of artificial general intelligence (AGI) ([30]). This pre-trained model shows a capacity for cross-domain task transfer, and its context understanding and creative output are close to the human level.

It is noteworthy that the concept of superintelligence (ASI) proposed by [5] ([5]) is any kind of intelligence that surpasses human cognitive ability in almost all fields. This concept is shifting from philosophical speculation to technical discussion. Although it is still in the stage of theoretical deduction, the integration of neuromorphic computing and quantum computing will accelerate this process.

### 2.3. Impact of Artificial Intelligence on Spectator Behavior

Against the background of the deep penetration of digital technology into the sports industry, the audience’s watching behavior is undergoing structural changes. The popularity of social media has changed the way sports organizations connect with their audiences. Artificial intelligence delves into social media platforms to extract valuable insights from user behavior patterns. By analyzing likes, shares, comments, and other interactions, it can fully understand the emotions and preferences of fans ([1]). Social media analysis has been upgraded from simple interactive measurement to multi-dimensional emotional analysis and promoting the platform to dynamically adjust its commentary strategy when the audience’s demand for technical analysis is detected, as well as real-time insertion of slow-motion replays and tactical diagrams ([3]). This technological breakthrough has led to the development of traditional demographic-based fan portraits into a dynamic model containing 230 dimensions of behavioral characteristics ([34]), which helps teams identify sub-groups with special preferences for behind-the-scenes content and thus customize their own experience and content.

By analyzing ten-year historical data, predictive modeling technology constructs an algorithm model with a spatiotemporal attention mechanism ([35]), which can accurately predict the corresponding trends of different groups in the release of sports content and commodities. By using more insightful data analysis methods, such as machine learning, deep learning, and knowledge graphs, it is possible to process large amounts of cross-temporal data, conduct complex calculations, and facilitate the aggregation and cross-validation of multi-dimensional, complementary datasets. The artificial intelligence algorithms can not only extract deep information and new knowledge from high-quality data but also digitize existing reliable experiences and knowledge into symbolic systems. Human knowledge and machine intelligence mutually confirm and complement each other, thereby enabling accurate prediction of the corresponding trends of different groups in sports content and product releases. This predictive ability enables sports organizations to realize the transformation from passive response to active guidance. In addition, a predictive algorithm can determine the likelihood that a viewer will be interested in an upcoming event, competition, or merchandise launch. This foresight allows teams to create targeted campaigns and promotional materials, ensuring that fans are exposed to content that matches their personal preferences ([26]). The interactive platform continuously optimizes the user interface through machine learning and dynamically adjusts content presentation according to the characteristics of individual behavior. For example, a sports app might learn that viewers are constantly checking live scores, watching highlights, and interacting with fantasy sports features. The application can then prioritize these elements, creating a personalized homepage that caters to the specific interests of the audience. This not only enhances the user experience but also encourages long-term interaction with the platform ([4]).

The integration of blockchain and artificial intelligence is reshaping the spectator economy, and some sports organizations have successfully adopted data-driven strategies. The NBA Top Shot platform uses blockchain technology to transform wonderful moments of the event into digital collections, creatively connecting the audience’s collective enthusiasm and experience. Similarly, Liverpool Football Club used machine learning to analyze the digital footprint of the audience and achieve precise customization of communication content, which increased the renewal rate of season tickets by 28% ([4]).

The evolution of technology has given birth to a new paradigm of audience behavior; the affective computing algorithm makes audience emotion the core parameter of content production, predictive model promotes behavior analysis from descriptive to predictive, and immersive interactive platforms reconstruct the connection between audience and club. This shift validates [9] ([9]) assertion that digital technology is building a two-way enabling mechanism between teams and spectators, enabling spectators to watch games from one-way consumption to an ecosystem of value co-creation.

The development of AI has undergone several key stages, from early symbolic reasoning to the current data-driven era dominated by deep learning. This paradigm shift, marked by breakthroughs like AlexNet, has enabled sophisticated applications, such as personalized recommendations and real-time data analysis, that are now transforming the sports industry and forming the basis of the modern spectating experience.

### 2.4. Social Identity Theory and Player Identity

#### 2.4.1. Overview of Social Identity Theory

Social identity is when a person believes that he or she identifies with a group, has an emotional bond with the group, and has the same or very similar characteristics, interests, or associations with other members of the group ([37]). Social identity theory suggests that individuals have both a personal identity, which includes characteristic attributes, such as abilities and interests, and a social identity, which consists of important group categories based on population classification (e.g., gender and race) or organizational membership (e.g., religion, education, and social organization) ([14]). Group members are functionalized in such a way that they are part of certain social groups ([7]). That is, social identity refers to an individual’s psychological commitment and sense of belonging to the group to which they belong ([39]). By virtue of this commitment and sense of belonging within the group, individuals can identify with the group.

Social identity theory explains that individuals tend to “satisfy their needs for positive self-esteem, a sense of belonging, a sense of control, and a holistic life by identifying with a particular group” ([46]). It is believed that group identity helps individuals maintain a degree of certainty about their high self-esteem ([36]).

In the field of sports management, social identity theory is widely used to study the phenomenon of audience identification with athletes ([24]). Player identification specifically refers to the psychological or emotional connection between fans and players. Fans establish an attachment relationship with players by emphasizing the commonalities between themselves and the players. If an audience member considers a star tennis player’s hitting style to be elegant and noble, he or she will try to establish an attachment relationship with the player through various means, including but not limited to wearing clothes, shoes, or hats with the player’s name or symbol and using products endorsed by the player. In the context of sports spectatorship, identification can manifest at various levels. While specific player identification focuses on a single star athlete ([8]), this study adopts the perspective of General Player Identification (GPI), which is rooted in social identity theory (SIT). GPI refers to the psychological state of perceiving oneself as a member of the social category “tennis players on the court” ([41]). This definition is crucial because, unlike identification with a single idol, GPI represents a broader, more stable connection to the athlete community as a whole, which is distinct from mere love for the sport ([18]). In this context, spectators high in GPI are motivated by the desire to maintain and enhance the positive distinctiveness of the athlete group.

Regarding the moderating effect of player identification, WAKEFIELD’s research suggests that even if the team’s performance is poor and the viewing experience is not good, fans with high identification tend to support their favorite team and will not reduce their emotional connection with the team due to its losses. Player identification and team identification have a high degree of homogeneity, so player identification may play a moderating role between artificial intelligence and viewing behavior.

#### 2.4.2. The Impact of Artificial Intelligence on Spectator Player Identity

The introduction of artificial intelligence technology provides a new way to shape and maintain the identity of sports audience groups but also brings many complex challenges. [43] ([43]) found that through the study of KEEP users of sports software, users have formed a significant sense of social identity in their interaction with AI assistants and community members. Specifically, AI assistants enhance users’ participation and sense of belonging through personalized feedback and community recommendation algorithms. This strong sense of social identity, in turn, encourages users to increase their interaction with the community, other users, and AI assistants in sports APPs, thus improving user stickiness.

[47] ([47]) focuses on social media platforms and explores the role of AI in group identity management with the help of text mining and sentiment analysis. Taking the comments on a college basketball team in the United States as the research object, AI technology can efficiently classify positive and negative emotions and formulate intervention strategies accordingly. For example, when the system detects a surge in negative comments, it automatically pushes the highlights of the team’s history, effectively turning negative emotions into positive interactions. This study shows how AI plays an important role in group identity, enhancing the audience’s emotional experience through emotional management and information guidance.

[20] ([20]) found that the existence and influence of others and the dissemination of social information can significantly affect the sense of similarity and distinction between individuals and others through the study of social identity and its role in artificial intelligence systems, especially in information sharing. This shows that in the process of shaping social identity, AI can effectively promote the interaction and information flow inside and outside of the group, thus affecting the identity structure of the audience group.

However, technological interventions do not always have a positive impact. [42] ([42]) conducted an online survey of 320 viewers, which showed that although most viewers had high trust in the accuracy, effectiveness, and fairness of artificial intelligence, such as a video assistant referee (VAR), viewers who had a higher sense of identity with their favorite team were less satisfied with VAR. The reason is that VAR removes the pleasure that spectators get from debating controversial decisions regarding their favorite athletes during the game, weakening the emotional value of the game and thus affecting the sense of identity between the audience and the team.

Based on the existing research, it can be seen that the influence of artificial intelligence on the audience identity of the event takes the following forms: on the one hand, algorithm-driven personalized interaction, emotional management, and community operation significantly improve the efficiency of audience participation and enhance the audience’s sense of belonging. On the other hand, the penetration of technical rationality in sports scenes may weaken the emotional value of sports as cultural rituals. Future research needs to explore more deeply how to retain the emotional elements that are difficult to quantify in sports culture while improving management efficiency.

### 2.5. Self-Determination Theory (SDT) and Spectator Motivation

#### 2.5.1. Overview of Self-Determination Theory

Self-determination theory (SDT) is a humanism-based motivation theory, which holds that human beings will develop in a more self-determined and autonomous direction in an appropriate social environment. [11] ([11]) proposed that social context factors are able to modulate individual motivation by influencing three basic, innate psychological needs: autonomy, competence, and belonging. When these three needs are satisfied, the individual’s motivation tends to be autonomously driven, but when these needs are hindered, the individual’s motivation tends to be externally controlled.

In recent years, the focus of SDT research has shifted from the simple distinction between intrinsic and extrinsic motivation to the distinction between autonomous and controlled motivation types ([11]). Autonomous motivation mainly comes from intrinsic motivation, integrative regulation, and certain regulation, among which intrinsic motivation is manifested in the high interest and pleasure of the behavior itself, integrative regulation means that the behavior has been consistent with the individual’s self-concept and values, and certain regulation means that the results of the individual’s behavior are consistent with the individual’s values. Controlled motivation, on the other hand, is at the low end of the continuum of self-determination and regulated by internal and external pressures, including introspective regulation and external regulation. Introspective regulation is usually manifested in behaviors triggered by guilt, shame, or the pursuit of self-worth, while external regulation is usually driven by external rewards, punishments, or threats ([33]).

Self-determination theory (SDT) also posits that human motivation is driven by the fulfillment of three innate psychological needs: autonomy (the desire to feel volitional and in control), competence (the desire to feel effective), and relatedness (the desire to feel connected to others) ([10]). Applying SDT to technology use, researchers increasingly view digital tools and platforms as a new social–environmental context that either supports or thwarts these needs ([17]). For AI-enabled sports viewing, the perceived features of AI (such as personalization, instant data, and community creation) act as proximal social factors that directly impact spectators’ need satisfaction and, consequently, their viewing motivation.

In the sports viewing context, specific AI functionalities map systematically onto the three SDT needs. Autonomy is satisfied by features that grant spectators personalized choice and interactive control (e.g., customized content streams); competence is fulfilled by AI’s ability to provide clear, real-time data and tactical analysis, enhancing the viewer’s mastery and understanding; and relatedness is supported by AI-driven social matching and community tools that foster a sense of belonging and shared experience.

Applied research in sports shows that compared with controlled regulation, autonomous regulation can often bring more positive and adaptive results, such as enhancing individual efforts, perseverance, performance, and mental health ([38]). These studies emphasize the importance of self-determination theory in promoting audience motivation and behavior, especially because when the audience can feel autonomy and intrinsic motivation, they tend to show higher motivation needs, satisfaction, and more sports participation.

#### 2.5.2. The Impact of Artificial Intelligence on Audience Motivational Needs

According to self-determination theory (SDT), when behavior is more autonomous and based on self-determined motivation, individual satisfaction increases significantly [27] ([27]). In the audience of sports events, the three basic needs of autonomy, competence, and belonging are also very important. By enhancing autonomy, enhancing the sense of competence, and promoting the sense of belonging, we can effectively meet the needs of the audience’s motivation to watch the game, thereby improving their satisfaction and participation.

AI technology can help viewers find more interesting competitions or activities through personalized recommendation systems. For example, intelligent assistants can provide users with detailed game statistics, player performance evaluation, and other information to help viewers understand the game more comprehensively. [45] ([45]) proposed that compared to traditional event referees, the human–machine collaborative sports event refereeing model involving artificial intelligence can convert abstract rules into intuitive images, enabling the audience to understand the basis for the decisions without relying on the commentary. AI can help identify and match users with similar interests, making it easier for viewers to find like-minded friends. [40] ([40]) believe that in sports communication, social media relationship chains can be introduced into AI spectator viewing scenarios. Through various forms, such as virtual communities and new media platforms, users can interact and communicate with others while watching the game, thereby establishing stronger social connections and enhancing their sense of belonging.

To sum up, the application of AI technology in the field of sports not only changes the traditional mode of watching games but also, more importantly, greatly improves participation motivation, satisfaction, and happiness by meeting the basic psychological needs of the audience: autonomy, competence, and sense of belonging. Specifically, AI features satisfy needs as follows.

AI and Autonomy: Personalization and interactive features (e.g., polling, customized replays) offer the viewer greater choice and control over the consumption path, thus satisfying autonomy.

AI and Competence: Real-time analysis and clear data visualization (e.g., probability metrics, tactical overlays) reduce the cognitive load and enhance the viewer’s understanding and mastery of the game, satisfying competence.

AI and Relatedness: AI-driven community tools and social matching functionalities foster a sense of belonging and shared experience with other fans, satisfying relatedness.

## 3. Research Hypothesis

### 3.1. The Relationship Between Artificial Intelligence, Spectator Motivation, and Spectator Behavior

Based on self-determination theory (SDT), when an individual’s behavior is more autonomous and motivated by self-determination, their satisfaction will be significantly improved. Autonomy, competence, and belonging, as the three basic psychological needs of human beings, are equally critical for the audience group of sports events, and meeting these needs can effectively improve the audience’s satisfaction and participation. The application of AI technology in the field of sports aims to meet these three needs systematically and to improve the motivation, satisfaction, and happiness of the audience by satisfying the audience’s autonomy, competence, and sense of belonging. Based on this, the research hypothesis H1 is proposed: Perceived AI-enabled spectating experience has a significant positive impact on viewing motivation (SDT Needs Satisfaction).

Second is the relationship between the spectator’s motivation and the intention of watching the game. When the audience’s motivation to watch the game is satisfied, their satisfaction will be significantly improved and further affect future behavior intention, including willingness to watch the game again and willingness to recommend. Based on this, the research hypothesis H2a is as follows: Audience viewing motivation (SDT Needs Satisfaction) has a significant positive impact on the willingness to watch the game. H2b: Audience viewing motivation (SDT Needs Satisfaction) has a significant positive impact on the willingness to recommend the game.

Finally, in terms of the relationship between artificial intelligence and the intention of watching the game, the popularity of social media has changed the way sports organizations connect with spectators. Artificial intelligence studies social media platforms in depth, extracts valuable insights from user behavior patterns, and comprehensively understands fans’ emotions and preferences by analyzing praise, sharing, comments, and other interactions, thus affecting spectators’ behavioral intentions. Based on this, the research hypothesis H3a is as follows: Artificial intelligence has a significant positive impact on the willingness to watch the game. H3b: Artificial intelligence has a significant positive impact on the willingness to recommend the game.

### 3.2. The Mediating Effect of Motivation to Watch Games and the Moderating Effect of Players’ Identification

First is the mediating effect of the viewing motivation (SDT Needs Satisfaction). Because artificial intelligence may have an impact on the audience’s motivation to watch the game and behavioral intention, and the motivation to watch the game may have a significant positive impact on behavioral intention, this paper believes that the motivation to watch the game may play a mediating role between artificial intelligence and behavioral intention. Based on this, the research hypotheses H4a and H4b are proposed, hypothesizing that the audience’s motivation to watch the game has a mediating effect on the relationship between artificial intelligence and the willingness to watch the game and the audience’s motivation to watch the game has a mediating effect on the relationship between artificial intelligence and the willingness to recommend the game.

Second is the moderating role of player identity in the relationship between AI and behavioral intention. Studies have shown that even if the audience’s experience of watching the game is not high due to the poor performance of the team, fans with high recognition tend to support their favorite team more and will not reduce their emotional connection because of the team’s failure. Therefore, this paper argues that player identification may play a moderating role between AI and audience behavioral intentions. The model is shown in Figure 1. Based on this, the research hypothesis H5a is as follows: player identification plays a moderating role in the mediation model of artificial intelligence → motivation to watch the game → willingness to watch the game. H5b: player identification plays a moderating role in the mediation model of artificial intelligence → motivation to watch the game → willingness to recommend.

## 4. Research Methods

### 4.1. Data Collection

In order to improve the reliability of the results, the study collected 272 sample data from Chinese tennis fans online through the questionnaire star platform in May 2025. Participants are allowed to decide whether to join the survey or not. Samples are efficiently obtained through online and social media channels. [31] ([31]) pointed out that this strategy is the most suitable scheme for the implementation of digital surveys. In order to protect the privacy of the respondents, the front page of the questionnaire clearly states that the data are only used for academic research and that the survey does not collect any personal information.

### 4.2. Variable Design

#### 4.2.1. Dependent Variable: Match Watching Behavior

The dependent variable of this paper is the spectator’s watching behavior. The measurement dimension is based on the theory of sports consumption behavior, which involves the willingness to participate directly and the tendency of social communication. Specifically, the willingness to watch is measured by questions such as “I plan to watch more tennis matches,” while the willingness to recommend is measured by questions such as “I am willing to recommend and invite others to watch tennis matches”. All observation items were measured using the Likert 5-scale, i.e., 1 = strongly disapprove, 2 = somewhat disapprove, 3 = fair, 4 = somewhat approve, and 5 = strongly approve.

#### 4.2.2. Independent Variable: Perceived AI-Enabled Spectating Experience

In this study, the independent variable is the perceived AI-enabled spectating experience, defined as a spectator’s overall perception of the usefulness and effectiveness of AI-driven features during sports events. Drawing from the existing literature on technology adoption and user experience in sports, this construct is operationalized through three key dimensions: (1) personalized recommendations, which reflect the degree to which AI provides tailored content (e.g., highlights, news); (2) information clarity, which assesses how AI tools (e.g., data visualization, Hawk-Eye) enhance the understanding of the game; and (3) interaction participation, which measures the extent to which AI facilitates social connections among fans (e.g., matching users with similar interests in virtual communities). Specifically, this article designs measurement questions such as “I think AI technology (such as personalized recommendations and game data analysis) can improve my viewing experience,” “I think artificial intelligence technology makes me better understand sports events,” and “AI can help me match viewers with similar interests”. All observation items are measured using the Likert 5-point scale, where 1 = strongly disapprove, 2 = somewhat disapprove, 3 = general, 4 = somewhat approve, 5 = strongly approve, and the average score of the question is taken as the final score of the independent variable of artificial intelligence.

#### 4.2.3. Moderating Variable: Player Identification

To minimize survey length, this study employed a single-item measure for player identification: “I have a sense of identity with tennis players on the court”. The use of a single-item measure for General Player Identification (GPI) warrants justification. While acknowledging the general preference for multi-item scales for complex constructs, single-item measures are validated and accepted under specific circumstances in behavioral science research ([2]). Specifically, when the construct is unambiguous and represents a single, clear, concrete concept, a well-phrased single item can achieve high face and predictive validity. Given our definition of GPI as the sense of psychological connection to the “group of athletes,” the item “I have a sense of identity with tennis players on the court” is clear and captures the core of group membership as theorized by SIT. This strategic choice also minimized the potential for participant fatigue and ensured high response quality across the broader survey.

#### 4.2.4. Mediating Variable: Motivation to Watch the Game

Self-determination theory holds that when the three psychological needs of autonomy, competence, and belonging are met, individuals will be more motivated to change or maintain their behavior. This paper argues that the experience of intelligent watching through AI + sports can affect the individual’s autonomy, sense of competence, and sense of belonging, thus further affecting their intention to watch the game. Mediating Variable: Viewing Motivation (SDT Needs Satisfaction). The viewing motivation scale was adapted from [Cite original scale source] using three items, each grounded in the fulfillment of the three basic psychological needs of SDT:

1. Autonomy: “I enjoy the opportunity to freely express my opinions and viewpoints about the game.” This item reflects the volitional control and self-direction enabled by AI’s interactive and personalized features.

2. Competence: “I feel I gain a deeper understanding of the match strategies through the information provided by AI.” This item assesses the spectator’s enhanced sense of efficacy and mastery over complex game content.

3. Relatedness: “I feel connected to other fans and experience resonance within the viewing community. This item measures the social belonging and emotional connection facilitated by AI-driven community features. Although three distinct dimensions of SDT (autonomy, competence, and relatedness) were conceptually mapped to guide item construction, the subsequent factor analysis confirmed that these needs converge to form a single, robust, higher-order construct, overall viewing motivation, indicating that spectators’ motivation is holistic when responding to the perceived AI-enabled environment. Therefore, in this paper, the motivation for watching the game is regarded as the mediating variable. The following factors are measured: ability to freely express my opinions and viewpoints on the tennis match (autonomy), sense of acquiring new knowledge and skills while watching the tennis match (competence), and sense of resonance of belonging in relevant tennis event communities or on social media (belongingness). All observation items were measured using the Likert 5-scale, i.e., 1 = strongly disapprove, 2 = somewhat disapprove, 3 = fair, 4 = somewhat approve, and 5 = strongly approve. The average scores of the questions were aggregated to form a continuous variable of motivation to watch the game. In this study, the Cronbach’s alpha coefficient of the scale was 0. 900, and the KMO value was 0. 871, which passed Bartlett’s sphericity test (*p* < 0.001).

#### 4.2.5. Control Variables

Control variables for this paper include (1) gender (male = 1, female = 2), (2) age (under 18 = 1, 18-24 = 2, 24-35 = 3, 35-45 = 4, over 45 = 5), and (3) match watching frequency (almost no active watching = 1, low frequency (less than 12 matches per year) = 2, medium frequency (1-3 matches per month), high frequency (1 or more matches per week)).

### 4.3. Measurement Model Evaluation

The latent factors and the observed items are presented in Table 1.

We evaluated the measurement model using Confirmatory Factor Analysis in AMOS 24.0 software. The model consists of four latent variables: perceived AI-enabled spectating experience, viewing motivation, viewing intention, and recommendation intention. The results of the Confirmatory Factor Analysis showed that the model had a good fit with the data; the ratio of chi-square to degrees of freedom (χ^2^/df) = 2.31, the comparative fit index (CFI) = 0.965, the goodness-of-fit index (GFI) = 0.913, the Tucker–Lewis index (TLI) = 0.956, and the root mean square error (RMR) = 0.037.

The internal consistency of the scale was verified, and the Cronbach’s alpha coefficient of each latent factor was above the critical value of 0.70. The convergent validity was established through the following criteria: the loading values of the observed variables of each latent factor were greater than 0.7, and the average variance extracted of each latent factor was higher than 0.5. The discriminant validity was also verified. According to the Fornell–Larcker Criterion, because the square root of the AVE value of each latent factor was greater than the correlation coefficient of that latent variable with any other latent variable, it was determined that the measurement model had good discriminant validity (Table 2 and Table 3).

### 4.4. Ethical Statement

Ethical review and approval were waived for this study in accordance with the local legislation and institutional requirements (Article 32 of Measures for Ethical Review of Life Sciences and Medical Research involving Human Beings of China; detailed information can be found at https://www.gov.cn/zhengce/zhengceku/2023-02/28/content_5743658.htm, accessed on 20 September 2023), as it did not entail clinical trials or manipulations involving humans or animals.

## 5. Results

### 5.1. Descriptive Statistics of Main Variables

Table 4 shows the descriptive statistical results of the variables. In the sample of this study, the mean value of the audience’s intention to watch the game is 3.665, the standard deviation is 1.067, the mean value of the recommendation intention is 3.813, and the standard deviation is 1.041, indicating that there are great differences in the audience’s intention to watch the game. The maximum value of AI is 25 and the minimum value is 5, which shows the difference in the audience’s perception of AI + intelligent experience. The mean value of player identification is 3.794, indicating that the majority of spectators have high player identification. The minimum value of spectator motivation is 6, and the standard deviation is 5.127, which indicates that there are obvious differences in spectator motivation among the samples.

### 5.2. The Mediating Effect of Spectators’ Motivation Between AI and Willingness to Watch

In this paper, Model 4 of PROCESS was used to analyze the mediating effect, and the Bootstrap method was used to extract 5000 Bootstrap samples to estimate the 95% confidence interval of the mediating effect. The results are shown in Table 5 and Table 6. Artificial intelligence has an indirect effect of 0.0479 on the audience’s motivation to watch the game and on the willingness to watch the game, which reaches a significant level, indicating that there is a mediating effect. In addition, because the direct effect of AI on the behavior of watching matches is 0.0680, which also reaches a significant level, motivation to watch matches plays a partial mediating role between AI and willingness to watch matches, which supports the hypothesis H4a.

### 5.3. The Mediating Effect of Spectator Motivation Between AI and Recommendation Intention

Similar to the method of testing the mediating effect of audience motivation between artificial intelligence and willingness to watch, this paper uses SPSS 26.0 macro PROCESS to test the mediating effect of audience motivation between artificial intelligence and willingness to recommend. Table 6 and Table 7 show that the mediating effect of motivation to watch games between AI and recommendation intention is 0.0548, and it reaches a significant level. In this mediation model, the total effect of AI on recommendation intention is 0.1126, and the direct effect is 0.0579, both of which are statistically significant. Therefore, viewing motivation (SDT Needs Satisfaction) plays a partial mediating role between AI and recommendation intention, which supports hypothesis H4b.

### 5.4. Moderated Mediator Model Testing

Moderating effect analysis aims to analyze the interaction effect of independent variables and moderating variables on dependent variables when the independent variables and dependent variables are determined. In this paper, multiple regression analysis was performed using model 59 in the PROCESS program mentioned by Hayes, and the results are shown in Table 8. The interaction of spectator motivation and player identification significantly affected spectator’s willingness to watch the match (β = 0.2809, SE = 0.1035, t = 2.72, *p* < 0.01, CI = [0.1102,0.4517]) and willingness to recommend (β = 0.1621, SE = 0.2207, t = 3.21, *p* < 0.01, CI = [0.3451,1.0737]). The interaction coefficient between the motivation to watch the game and the level of player identification was 0.2809 (*p* < 0.01), indicating that the higher the level of player identification, the stronger the positive impact of the motivation to watch the game on the willingness to watch; conversely, the lower the level of identification, the weaker the influence of this path. The same is true for the influence on the willingness to recommend. The interaction coefficient between the motivation to watch the game and player identification was 0.1621 (*p* < 0.01), further verifying that the efficiency of motivation to convert into recommendation behavior is higher for those with high identification, while it is lower for those with low identification.

This suggests that player identification significantly moderates the relationship between spectator motivation and spectator behavior. Although there is a significant path relationship between the motives for watching the game and the audience’s viewing behavior, the magnitude of the path relationship varies with the degree of player identification.

Specifically, based on social identity theory and self-determination theory, this significant interaction effect suggests that the process of converting viewing motivation (SDT Needs Satisfaction) into behavior differs between high- and low-identification fans. A possible theoretical explanation for this phenomenon lies in contrasting decision making frameworks. For high-identification spectators, whose self-concept is deeply intertwined with the players, viewing behavior is an expression of identity. Thus, AI-enhanced motivation is more readily translated into action, a process consistent with an “identity consistency” model. In contrast, low-identification spectators may rely more on immediate experiential factors. Their motivation-to-behavior path appears more volatile, aligning with a “cost-benefit” rational decision model where external stimuli are continually needed to prompt behavioral input and the information obtained through artificial intelligence technology is mainly used to strengthen the existing identity. Meanwhile, the audience with low identity relies on the “cost-benefit” rational decision making model and needs external stimuli to continuously promote their behavioral input. However, it should be noted that audiences from different regions and with different cultural backgrounds have different priorities when it comes to real-time perception during the event. Based on this finding, the optimization of an intelligent game watching system can start from the following perspectives: for the core fan group, the emotional connection function of AI should be strengthened, such as through AI-generated personalized historical recaps that support the team, as well as a fan community interaction module. For the long-tail users, dynamic content optimization strategies need to be designed based on regional sports culture characteristics. For example, for the Chinese market, real-time data interpretation of local athletes can be added, and, for the Southeast Asian market, localized language-based interactive games for events can be developed in order to compensate for the behavioral inertia caused by the lack of identity and to adapt to the cost–benefit decision preferences of low-identification audiences in different regions.

## 6. Discussion and Conclusions

This study empirically investigated the impact of the perceived AI-enabled spectating experience on the viewing behavior of sports fans, with a particular focus on the mediating role of viewing motivation (SDT Needs Satisfaction) and the moderating role of player identification. The research process and results have revealed several key behavioral trends that align with the initial hypotheses.

### 6.1. Key Findings

The main conclusions of this research are as follows.

The measurement models developed for the perceived AI-enabled spectating experience, viewing motivation (SDT Needs Satisfaction), and spectator behavior demonstrated good internal consistency, reliability, and validity. Given the scarcity of established scales in this specific domain within China, these instruments can provide a valuable reference for future research.

Viewing motivation (SDT Needs Satisfaction) plays a partial mediating role in the relationship between the perceived AI-enabled spectating experience and viewing behavior. The indirect effect of the perceived AI experience on both viewing intention and recommendation intention through viewing motivation (SDT Needs Satisfaction) was statistically significant. This indicates that while the AI experience has a direct effect on behavior, it also significantly influences behavior indirectly by satisfying spectators’ motivations.

Player identification positively moderates the relationship between viewing motivation (SDT Needs Satisfaction) and viewing behavior. For spectators with higher levels of player identification, their viewing motivation (SDT Needs Satisfaction) more strongly translates into both viewing intention and recommendation intention. However, player identification did not moderate the relationship between the perceived AI experience and viewing motivation.

### 6.2. Theoretical Implications

This result suggests that the perception of AI-enabled features is not merely a technical factor but acts as a significant antecedent to motivation. The perceived AI-enabled spectating experience can indirectly influence viewing behavior by satisfying spectators’ multi-dimensional needs.

This study contributes to self-determination theory (SDT) by demonstrating how perceptions of technology can act as a social context factor that influences basic psychological needs. The findings show that a positive AI-enabled experience fulfills spectators’ needs for autonomy, competence, and belongingness (conceptualized as viewing motivation), which in turn drives autonomous behavioral intentions like watching and recommending.

Furthermore, the research extends social identity theory (SIT) in the context of sports marketing. The finding that player identification strengthens the motivation–behavior pathway aligns with the core principle of SIT, which posits that identity consistency is a powerful driver of behavior. It highlights that for highly identified fans, viewing behavior is not just consumption but an affirmation of their social identity.

### 6.3. Practical Implications

The non-significant moderating effect of GPI on the AI→Motivation path offers a crucial theoretical insight: the perception of AI’s utility is fundamentally objective and universal, independent of social identity. Whether an AI feature provides clear data or effective personalization is a matter of functional effectiveness, which is assessed similarly by both high- and low-GPI fans. Therefore, AI’s ability to satisfy basic psychological needs is intrinsic to the technology’s design, not the spectator’s identity. The role of identity is thus reserved for the subsequent, more volitional stage of behavioral conversion, acting as a powerful filter ([18]) that selectively amplifies motivated behavior based on identity maintenance needs.

To effectively enhance audience engagement and consumption, event managers should focus on the strategic application of AI technology to cultivate spectator motivation. Beyond uncontrollable factors like player performance, managers should leverage AI to continuously improve auxiliary products and services to meet the diverse needs of the audience. For instance, AI-driven analysis of audience data can optimize stadiums’ functional design. Advanced audiovisual systems powered by AI can create more immersive viewing effects, while AI algorithms can facilitate personalized promotions and midfield interactions to stimulate viewing motivation (SDT Needs Satisfaction).

A differentiated marketing strategy based on player identification is crucial. For fans with high player identification, marketers should use AI’s deep analysis of their emotional preferences to strengthen the fan–player connection, enhancing their emotional attachment and loyalty. For fans with low player identification, the focus should be on using AI to accurately identify service-related pain points and improve service quality, thereby enhancing satisfaction and promoting viewing behaviors.

### 6.4. Limitations and Future Research

This study has several limitations that offer avenues for future research. The first limitation concerns the single-item measure used for General Player Identification (GPI). Although we provided a substantive justification in the Methodology section (Section 4.2.3, lines 525–535), arguing that the item is conceptually clear and supported by the methodological literature ([2]), we must acknowledge that its use presents a trade-off. While it ensured high response rates and reduced participant fatigue, a multi-item scale would offer greater reliability and allow for a more nuanced distinction between GPI and specific idol identification. Future research should employ validated multi-item scales to capture different facets of player identification—for example, measuring emotional commitment and cognitive awareness separately—to further explore the boundaries of the moderation effect found in this study. This comparative approach would enhance the robustness of the GPI construct in the context of AI-enabled spectatorship. Secondly, the measurement of player identification relied on a single-item scale, which may not fully capture the complexity of the construct and is susceptible to measurement error. Future studies should employ validated, multi-item scales to provide a more robust test of the moderating effect. Thirdly, our independent variable, the perceived AI-enabled spectating experience, was based on self-reported perceptions rather than actual interaction with specific AI systems. Future research could utilize experimental designs to assess the causal impact of different AI features on spectator behavior. Finally, the study was conducted on a sample of Chinese tennis fans, which may limit the generalizability of the findings to other sports and cultural contexts. Cross-cultural comparative studies are needed to validate the proposed model.

## Figures and Tables

**Figure 1 behavsci-15-01702-f001:**
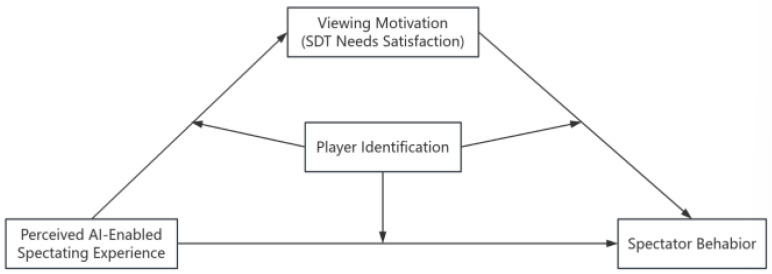
Study hypothesis model.

**Table 1 behavsci-15-01702-t001:** Factors and observed variables.

Factor Name	Observed Variable
Perceived AI-Enabled Spectating Experience (EXP)	1-1 I think AI technology (such as personalized recommendations and game data analysis) can improve my viewing experience (EXP1)
	1-2 I think artificial intelligence technology makes me better understand sports events (EXP2)
	1-3 AI can help me match viewers with similar interests (EXP3)
	1-4 The AI-driven interactive features (such as virtual communities, real-time win rate predictions) have enhanced my interaction with the players in the event. (EXP4)
Viewing Motivation (SDT Needs Satisfaction) (MOT)	2-1 My ability to freely express my opinions and viewpoints on the tennis match (MOT1)
	2-2 The sense of acquiring new knowledge and skills while watching the tennis match (MOT2)
	2-3 The sense of resonance I can obtain in the relevant tennis event communities or on social media (MOT3)
The Willingness To Watch (WAT)	3-1 I am willing to watch the game again (WAH1)
	3-2 I am willing to participate in events related to the game (WAH2)
	3-3 I am willing to purchase products related to the game (WAH3)
The Willingness To Recommend (REC)	4-1 I am willing to recommend and invite others to watch the game (REC1)
	4-2 I am willing to share information about the game with others (REC2)

**Table 2 behavsci-15-01702-t002:** Psychometric properties of measurement models.

Latent Construct	Observed Item	Factor Loading	Cronbach’s α	Average Variance Extracted	CR
Perceived AI-Enabled Spectating Experience	EXP1	0.789	0.925	0.648	0.880
	EXP2	0.829			
	EXP3	0.793			
	EXP4	0.808			
Viewing Motivation (SDT Needs Satisfaction)	MOT1	0.754	0.900	0.588	0.811
	MOT2	0.778			
	MOT3	0.769			
The Willingness to Watch	WAT1	0.834	0.903	0.692	0.875
	WAT2	0.812			
	WAT3	0.849			
The Willingness to Recommend	REC1	0.807	0.875	0.637	0.778
	REC2	0.790			

**Table 3 behavsci-15-01702-t003:** Square root of construct’s AVE and its correlation with any other construct.

	Perceived AI-Enabled Spectating Experience	Motivation to Watch the Game	The Willingness to Watch	The Willingness to Recommend
Perceived AI-Enabled Spectating Experience	0.805			
Viewing Motivation (SDT Needs Satisfaction)	0.634	0.767		
The Willingness to Watch	0.582	0.615	0.832	
The Willingness to Recommend	0.557	0.532	0.658	0.798

**Table 4 behavsci-15-01702-t004:** Descriptive statistical results of main variables.

Variable Name	Sample Size	Mean	Median	Standard Deviation	Minimum	Maximum
Willingness to watch the game	272	3.665	4	1.067	1	5
Willingness to recommend	272	3.813	4	1.041	1	5
Perceived AI experience	272	18.22	18	4.452	5	25
Player identity	272	3.794	4	1.010	1	5
Viewing motivation (SDT Needs Satisfaction)	272	22.11	23	5.127	6	30
Age	272	2.886	2	1.360	1	6
Gender	272	1.493	1	0.501	1	2
Frequency of watching games	272	2.607	3	1.189	1	4

**Table 5 behavsci-15-01702-t005:** The mediating effect of watching motivation between perceived AI experience and watching intention.

	Dependent Variable: Motivation to Watch the Game				Dependent Variable: Willingness to Watch the Game			
Variable Name	β	SE	t	95%CI	β	SE	t	95%CI
Perceived AI experience	0.1471	0.0100	14.68 ***	[0.1274,0.1668]	0.0680 0.3259	0.0154 0.0698	4.43 ***	[0.0378,0.0982] [0.1885,0.4633]
Viewing motivation (SDT Needs)							4.67 ***	
R^2^			0.4602				0.3989	
F			56.91 ***				35.30 ***	

*** Symbol indicates *p* < 0.001, indicating an extremely significant difference.

**Table 6 behavsci-15-01702-t006:** The mediating effect of viewing motivation between AI and watching intention.

	Dependent Variable: Motivation to Watch the Game				Dependent Variable: Willingness to Recommend			
Variable Name	β	SE	t	95%CI	β	SE	t	95%CI
Perceived AI experience	0.1471	0.0100	14.68 ***	[0.1274,0.1668]	0.0579 0.3724	0.0156 0.0708	3.72 ***	[0.0272,0.0885] [0.2331,0.5117]
Viewing motivation (SDT Needs)							5.26 ***	
R^2^			0.4602				0.3497	
F			56.91 ***				28.61 ***	

*** Symbol indicates *p* < 0.001, indicating an extremely significant difference.

**Table 7 behavsci-15-01702-t007:** Indirect, direct, and total effects.

	Direct Effect	Indirect Effects	Total Effect	Hypothesis Verification
Perceived AI experience → motivation to watch the game → willingness to recommend	0.0579 ***	0.0548 ***	0.1126 ***	H4b is established

*** Symbol indicates *p* < 0.001, indicating an extremely significant difference.

**Table 8 behavsci-15-01702-t008:** Analysis of mediated mediating effect.

	**Dependent Variable: Motivation to Watch the Game**				**Dependent Variable: Willingness to Watch the Game**			
**Variable Name**	β	**SE**	**t**	**95%CI**	β	**SE**	**t**	**95%CI**
Perceived AI experience	0.0910	0.0279	3.26 **	[0.0360,0.1460]	0.0425	0.0175	2.43 **	[0.0247,0.0977]
Player identity	0.3149	0.1329	2.37 **	[0.0532,0.5766]	0.2825	0.1511	1.87 *	[0.2030,0.3620]
AI × player identity	0.0020	0.0072	0.28	[−0.0121,0.0162]	0.0429	0.0620	0.69	[0.0595,0.1453]
Viewing motivation (SDT)					0.7094	0.2207	3.21 ***	[0.3451,1.0737]
Motivation × player identity					0.2809	0.1035	2.72 ***	[0.1102,0.4517]
R^2^			0.5377				0.4070	
F			51.38 ***				22.56 ***	
	**Dependent Variable: Motivation to Watch the Game**				**Dependent Variable: Willingness to Recommend**			
**Variable Name**	β	**SE**	**t**	**95%CI**	β	**SE**	**t**	**95%CI**
Perceived AI experience	0.0767	0.0202	3.80 ***	[0.0434,0.1099]	0.1740	0.1042	1.67 *	[0.0020,0.3459]
Player identity	0.1626	0.0959	1.70 *	[0.0044,0.3209]	0.1381	0.0744	1.85 *	[0.0499,0.4260]
AI × player identity	0.0081	0.0052	1.56	[−0.0005,0.0166]	0.0116	0.0121	0.96	[0.0316,0.0084]
Viewing motivation (SDT)					0.5841	0.2222	2.62 ***	[0.2173,0.9509]
Motivation × player identity					0.1621	0.0485	3.34 ***	[0.0821,0.2422]
R^2^			0.6774				0.4436	
F			65.72 ***				26.21 ***	

*** Symbol indicates *p* < 0.001, indicating an extremely significant difference, the ** symbol repre-sents *p* < 0.01, indicating a relatively significant difference, and the * symbol indicates *p* < 0.05, in-dicating a significant difference.

## Data Availability

The data are not publicly available due to privacy restrictions, as they include responses from human participants.

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
