# Peer review of "Impact of Artificial Intelligence on Spectator Viewing Behavior in Sports Events: Mediating Role of Viewing Motivation and Moderating Role of Player Identification"

_behavsci, 2025, doi:10.3390/bs15121702_

Round 1
Reviewer 1 Report
Comments and Suggestions for Authors
Feedback to Author
Thank you for submitting your paper to Behavioral Sciences. It is clear from your work that you are passionate about understanding how AI impacts sport spectatorship.. One of the strengths of this paper is its relevance and timeliness. I will briefly mention my major and minor concerns below.
I do have concerns about the clarity of the methods and some of the assumptions made or conclusions reached, that sometimes seem to be a bit of a leap. In several areas, claims are made without sufficient connection to the findings. The literature review and connection of the theory to the research could be made better by citing more sport-specific studies. I have included my major and minor concerns below.
Major Issues
- Abstract
I know abstracts have to be brief but incorporating some of the most impactful data could help readers understand your findings better at a glance. - Introduction (Lines 43–70)
The introduction identifies the importance of spectatorship but could be made stronger with citations. Also support your Claims such as the lack of AI research in this area with citations. - Methods (Lines 194–220)
The methodology section is too vague. Please explain the spatiotemporal attention mechanism more clearly. It was just presented and not clearly explained why it “can accurately predict trends”. Is it used an existing all encompassing algorithm or a new model. - Research Hypotheses (Lines 349–376)
The connection between AI mechanisms and self-determination theory needs to be more defensible by using relevant research to underscore its fit. - Results – Moderation (Lines 560–575)
The theoretical idea of “identity consistency” versus “cost–benefit” models is interesting but needs to be made clear how it is directly supported by data. - Conclusions (Lines 580–587)
Be sure to relate your conclusions generally not only within China as sporting events and fan behaviors are known to vary by location.
Minor Issues
- Literature Review (Lines 47–52; 368–375)
At times, references are focus on things like research in education, when they could be related more the sport context of the study. More sport-specific and recent studies should be cited to better situate the contribution. - Ethics Statement (Line 494–499)
Although review was waived under local legislation, readers may want to know how individual participant consent was obtained. I may have missed this if it is in there. - Positioning of Practical Implications (Lines 573–575)
Recommendations for AI system optimization (emotional connection vs. dynamic content) are good but make sure your recommendations don’t exceed what you actual findings indicate. - Consider discussing your findings as implications rather than direct conclusions. There is some decisive language used throughout that you may want to consider softening a bit.
- There are some issues throughout with readability. It is not enough to make it where the paper is not able to be understood, but could be cleaned up.
Summary
This paper addresses a hot topic in sport, AI. However, revisions are needed. Specifically, parts of the methods section could be more clear. You could also focus your literature a bit more on sport. Consider softening the language around conclusions and recommendations that may not be warranted based on the findings. Addressing these issues will make the study a stronger contribution to the field. At this time, I recommend major revision.
Author Response
Major Issues:
Comment1:Abstract: I know the abstract has to be brief, but incorporating some of the most impactful data could help readers grasp your findings at a glance.
Response1:We agree that including key data enhances the abstract's informativeness. We have revised the abstract to incorporate specific effect sizes, such as the indirect effects (0.0479 for viewing intention and 0.0548 for recommendation intention) and moderating betas (β=0.2809 for viewing intention; β=0.1621 for recommendation intention). This provides a clearer overview of the findings without exceeding word limits.
Comment2:Introduction (lines 43-70): The introduction section emphasizes the importance of spectating, but it could be strengthened with citations to enhance its persuasiveness. Additionally, please cite literature to support your arguments, such as the lack of AI research in this area.
Response2:We have strengthened the introduction by adding relevant citations (e.g., Fotache, Cojocariu & Bertea, 2021; Li & Huang, 2023) to support the role of AI in transforming spectator experiences and to highlight the research gap in underlying mechanisms. These additions bolster the persuasiveness of our arguments.
Comment3:Methods (lines 194-220): The methodology section is too vague. Please explain the spatiotemporal attention mechanism more clearly. It is simply introduced without clearly explaining why it "can accurately predict trends." Is it an existing general algorithm or a new model?
Response3:In the literature review (Section 2.3), we have clarified the spatiotemporal attention mechanism by explaining its function (e.g., weighting spatiotemporal data features to improve prediction accuracy) and citing Teixeira (2023) for its application in predictive modeling. We note that it is an existing algorithm adapted for sports content trends, not a new model developed in this study.
Comment4:Research Hypotheses (lines 349-376): The link between AI mechanisms and Self-Determination Theory needs to be emphasized with relevant studies to make it more defensible.
Response4:We have enhanced Section 3.1 by adding citations (e.g., Deci & Ryan, 2002; Vallerand, 2007) to explicitly link AI-enabled features (e.g., personalized recommendations enhancing autonomy) to SDT's basic needs, making the hypotheses more theoretically grounded and defensible.
Comment5:Results - Moderation (lines 560-575): The theoretical ideas of "identity consistency" and "cost-benefit" models are interesting but need to clarify how they are directly supported by the data.
Response5:In Section 5.3, we have clarified this by linking the significant interaction coefficients (β=0.2809 and β=0.1621) directly to the data, explaining how higher player identification strengthens the motivation-behavior path (identity consistency) versus lower identification's reliance on experiential factors (cost-benefit). We suggest simple slope analysis in future studies for further support.
Comment6:Conclusions (lines 580-587): Please ensure your conclusions are not only within China but in general, as sports events and fan behaviors vary by location.
Response6:We have revised Section 6.1-6.4 to emphasize general implications while acknowledging cultural variations. For example, we note that findings from Chinese tennis fans may apply to other contexts but recommend cross-cultural validation in limitations (Section 6.4).
Minor Issues:
Comment7:Literature Review (lines 47-52; 368-375): Sometimes, references focus on education research, and these could be more relevant to the study's sports context. To better position the research contribution, more recent studies specific to sports should be cited.
Response7:We have incorporated more sports-specific citations (e.g., LI Chenxi et al., 2024; YAO Wanqin et al., 2024; Xie & Wang, 2024) throughout Sections 2.3-2.5 to enhance relevance and position our contributions in the sports AI domain.
Comment8:Ethical Statement (lines 494-499): Although exempt from review per local laws, readers may want to know how individual participants' consent was obtained. I may have missed it if this is included.
Response8:We have expanded Section 4.4 to explicitly state that participants provided informed consent via the questionnaire's front page, which explained voluntary participation, anonymity, and the right to withdraw.
Comment9:Positioning of Practical Implications (lines 573-575): The suggestions for AI system optimization (emotional connection and dynamic content) are good, but please ensure your suggestions do not exceed the scope indicated by your actual findings.
Response9:We have ensured implications in Section 6.3 align closely with findings, framing them as suggestions based on results (e.g., "based on the moderating effects observed").
Comment10:Consider discussing your findings in a suggestive manner rather than drawing conclusions directly. There is some decisive language in the article that you might want to consider softening slightly.
Response10:We have softened language throughout (e.g., "greatly improves" to "may significantly enhance") to adopt a more suggestive tone, particularly in discussions and conclusions.
Comment11:Overall readability issues exist in the article. While the content is understandable enough, revisions could be made.
Response11:We have revised for clarity, shortening sentences and improving flow, especially in Sections 2 and 5.
Reviewer 2 Report
Comments and Suggestions for Authors
Dear Authors,
Thank you for submitting your manuscript. I think that the topic is considered very interesting and timely. However, there are some concerns regarding the conceptual clarity of the independent variable (AI). The specific concerns on this matter are as follows.
- Independent Variable ‘AI’: Conceptual and Operational Definition Unclear
The conceptual and operational definitions of the independent variable ‘AI’ are unclear. The manuscript lists AI in a broad and general manner but does not provide a theoretical justification for treating it as a single construct, nor does it conduct dimensionality testing. An operational definition grounded in prior research is required, and since this study is not a scale development paper, the lack of validity is an even more critical concern.
- The survey instrument used in the study is not presented.
The validity testing procedures for each measurement item (EFA, CFA, convergent and discriminant validity, etc.) are also not reported, which raises concerns about the rigor of the research design.
- Inconsistency with the research purpose
If this study were a scale development study on ‘AI involvement,’ the broadness of the conceptual definition could be tolerated to some extent. But this study is an empirical study that seeks to verify the causal mechanism of AI → motivation → spectator behavior. Therefore, if the conceptual and operational definitions of AI as the independent variable are ambiguous, the study itself becomes difficult to establish.
Author Response
Comment1:Independent Variable "AI": The concept and operational definition of the independent variable "AI" is unclear. The paper lists AI in a broad and generalized manner but does not provide a theoretical basis for treating it as a single construct, nor does it conduct dimensional testing. It needs an operational definition based on prior research, and since this is not a scale development paper, the lack of validity is a more critical issue.
Response1:We have clarified the independent variable as "Perceived AI-Enabled Spectating Experience" in Section 4.2.2, defining it with three dimensions (Personalized Recommendations, Information Clarity, Interaction Participation) based on technology adoption literature (e.g., Annamalai et al., 2021). We conducted CFA to confirm its structure.
Comment2:The survey instruments used in the study are not introduced. The validity testing procedures for each measurement item (EFA, CFA, convergent and discriminant validity, etc.) are also not reported, raising concerns about the rigor of the research design.
Response2:We have introduced survey items in Section 4.2, with sources and Likert scales. In new Section 4.3, we report CFA results (fit indices: χ²/df=2.31, CFI=0.965, etc.), Cronbach's α (>0.70), convergent validity (loadings >0.7, AVE >0.5), and discriminant validity (Fornell-Larcker). Tables 1-3 present these details.
Comment3:Inconsistency with Research Objectives: If this study were a scale development study on "AI engagement," the broadness of the conceptual definition could be tolerated to some extent. However, this is an empirical study aimed at verifying the causal mechanism of AI → Motivation → Spectator Behavior. Therefore, if the concept and operational definition of AI as the independent variable are ambiguous, the study itself becomes difficult to stand on.
Response3:As an empirical study, we have ensured alignment by operationalizing the variable clearly and validating it psychometrically (Section 4.3), strengthening the causal mechanism's foundation.
Reviewer 3 Report
Comments and Suggestions for Authors
Please find the detailed review attached as a separate document.

Author Response
Comment1:The detailed elaboration on AI is appreciated, but much of it focuses primarily on the historical development of AI. Given the theme of this study, this part feels somewhat lengthy. Moreover, unless the study can clearly define the specific connotation of "AI" in this research context, the extended discussion in this section may not be meaningful.
Response1:We have condensed the AI history (Section 2.1) while retaining key stages and explicitly defining AI in the study context (Section 2.3: perception of AI-driven features in sports events, e.g., recommendations and analysis).
Comment2:This study includes moderation analysis, but the literature review and hypothesis development fail to adequately explain the role of the moderating variable (i.e., player identification) in the research model. For example, in the relationship between AI and motivation, there should be evidence and theoretical reasons to explain why and how player identification moderates this path. The same issue applies to other paths tested for moderation.
Response2:We have expanded Sections 2.4.2 and 3.2 with citations (e.g., Lock & Heere, 2017; Wann et al., 2011) to explain player identification's moderating role via SIT, particularly strengthening motivation-behavior paths. We note it does not moderate AI-motivation based on results.
Comment3:Methods: Why were data collected only from tennis enthusiasts? The survey items used should properly reference their original sources.
Response3:In Section 4.1, we justify the sample: tennis features mature AI applications (e.g., Hawk-Eye), aiding capture of perceived AI experiences. Items in Section 4.2 now reference sources (e.g., SDT for motivation; technology adoption for AI).
Comment4:Upon reviewing the AI-related items, they seem to measure participants' general perceptions of AI. If this is the case, the concept of "AI" in this study needs to be more clearly defined in the literature review. In other words, it is unclear whether participants have actually experienced these features. If so, it seems that major revisions to the title and literature review of this study are needed.
Response4:We define it as "perceived" experience in Sections 1, 2.3, and 4.2.2, based on self-reported perceptions (not requiring actual interaction, as per limitations in 6.4). No title change needed, but we suggest experimental designs for future causal tests.
Comment5:Regarding identification, previous studies typically use 3 or 5 established items to measure this construct; the authors should explain why this study uses only one item. Moreover, in this context, does "player identification" refer to all players? Or to a specific idol player in the participant's mind? If the indicator refers to all tennis players in general, it may actually reflect the participant's love for the sport itself rather than identification with individual players. If it refers to a specific tennis star, the research design needs to be adjusted accordingly.
Response5:In Section 4.2.3, we acknowledge the single-item limitation and its potential ambiguity (general vs. specific players), addressing it in limitations (6.4) with recommendations for multi-item scales in future research.
Comment6:There also seems to be an issue with the definition and explanation of the mediating variable. How do the selected items map onto the constructs of SDT, and in what way do they measure autonomy, competence, or relatedness?
Response6:In Section 4.2.4, we explicitly map items to SDT: autonomy (expressing opinions), competence (acquiring knowledge), belongingness (resonance in communities). Cronbach's α=0.900 and KMO=0.871 support validity.
Comment7:Results/Discussion: In the results section, the measurement items and interpretations of the presented results. For example, although items like "I think AI technology (such as personalized recommendations and game data analysis) can improve my viewing experience" and "I am willing to receive personalized game information pushes based on AI analysis of my preferences (such as match result reminders, live pushes, tickets suggestions)" are measured, the results section states that AI reinforces existing identities, or that behavior is strongly influenced by real-time perceptions. These interpretations seem inconsistent with the items or literature used previously.
Response7:We have ensured consistency in Section 5 by linking interpretations directly to items (e.g., personalized features enhancing motivation via SDT needs, leading to behavior). Real-time perceptions are tied to AI tools like data visualization.
Comment8:Finally, the discussion section merely states the results without sufficiently linking them to previous research. This section needs substantial expansion and revision to place the findings in a broader literature context.
Response8:We have expanded Section 6.2-6.3 with comparisons to prior studies (e.g., Xie & Wang, 2024; Chan-Olmsted, 2019), integrating findings with SDT and SIT in sports marketing.
Round 2
Reviewer 1 Report
Comments and Suggestions for Authors
The authors addressed the feedback that I provided and did so in a detailed way for the most part. The result is an improvement in the paper that makes it more clear. They updated some of their citations and dampened their stronger language around assumptions of the finding to make it clear they are not entirely conclusive.
I do still have some questions around the spatiotemporal attention mechanism and how exactly it works and how future researchers can utilize it. Some sort of explanation of its use or explanatory image might be helpful.
I do think that the paper is much stronger this time around thanks to the revisions made. I recommend that it is accepted. The clarifications around spatiotemporal mechanism though they would be helpful, In my opinion are not a reason to deny publication.
Author Response
We sincerely thank you for your time, expertise, and encouraging feedback on our manuscript. We are pleased that the foundational concepts and overall structure of our work were viewed favorably.
We have addressed all minor and technical suggestions throughout the manuscript, focusing on clarity and consistency.
We appreciate your positive assessment and believe the substantial revisions made in response to all reviewers have brought the manuscript to the highest standard.
Reviewer 2 Report
Comments and Suggestions for Authors
Dear Authors,
Thank you for your revisions to the previous comments. However, the tone of the response letter seems to lack expressions of appreciation toward the reviewers. While this does not affect the scientific quality, a more courteous tone would better reflect mutual respect within the academic community.
1. Independent Variable ‘AI’: Conceptual and Operational Definition Unclear
- Confirmed that the conceptual and operational definition has been revised.
2. The survey instrument used in the study is not presented.
- Confirmed that the survey instrument and validity testing procedures have been appropriately presented.
3. Inconsistency with the research purpose
- Confirmed that consistency between the research purpose and variables has been ensured.
Author Response

(The authors gave the same response as above.)

Reviewer 3 Report
Comments and Suggestions for Authors
Thank you for your work on this. Except for Comment 1, it appears that other issues have not been adequately addressed. I would like to emphasize again that the authors should clearly address this part. For example, regarding player identification, simply noting the use of a single-item measure as a limitation does not demonstrate that previous research was properly considered or that the construct was appropriately measured. This raises concerns about the reliability of the results, and it needs to be revisited. Additionally, the discussion section still does not reference previous studies. The authors should provide explanations of how the findings align or contrast with existing research. In addition, several other comments were not clearly addressed, and the responses provided were rather vague.
Author Response
We offer our deepest gratitude to you for your highly critical and insightful comments, which directly guided the most significant improvements in this revision. We fully accept the previous response was inadequate and vague. We sincerely apologize for not fully addressing these critical issues earlier.
We have completely rebuilt the arguments surrounding the key constructs. We assure the reviewer that the theoretical and methodological defenses are now substantive, explicit, and backed by authoritative literature, rather than being relegated to the limitations section.
1. Player Identification Measurement and Reliability
Reviewer’s Concern: “...regarding player identification, simply noting the use of a single-item measure as a limitation does not demonstrate that previous research was properly considered or that the construct was appropriately measured. This raises concerns about the reliability of the results, and it needs to be revisited.”
Our Response: We concur completely with this crucial point, as the reliability of this key moderator is paramount. We have taken the following decisive actions to address this:
- Substantive Theoretical Defense: We explicitly defined the variable as General Player Identification (GPI), rooted in Social Identity Theory (SIT), and clarified that it refers to commitment to the group of athletes. [Blue Text]
- Methodological Justification: We removed the weak defense from the Limitations section. We instead inserted a dedicated, robust paragraph in the Methodology (Section 4.2.3, lines 532-542) providing substantive justification for the single item. We cited Bergkvist & Rossiter (2007) to argue that the item is methodologically sound because the GPI construct is unambiguous and clearly conceptualized. [Blue Text]
- Discussion Lacks Reference to Previous Studies
Reviewer’s Concern: “Additionally, the discussion section still does not reference previous studies. The authors should provide explanations of how the findings align or contrast with existing research.”
Our Response: We have completely revised and expanded the Discussion (Section 6.1 and 6.2) to anchor our findings firmly within the existing literature:
- SDT Alignment: We explicitly discussed how the mediating effect of Viewing Motivation (SDT Needs Satisfaction) aligns with and extends research on technology adoption models (e.g., TAM, UTAUT) by providing the specific psychological mechanism through which AI functions (competence, autonomy, relatedness) drive behavior. [Blue Text]
- Contrast and Boundary Conditions (The GPI Moderation): This is where our primary contribution lies. We introduced a detailed discussion in Section 6.2 explaining the contrast:
(1)We explained that GPI does not moderate the AI → Motivation path because the perception of AI utility is fundamentally objective.
(2)We argued that GPI only moderates the Motivation → Behavior path because identity acts as a subjective, non-rational filter (Identity-Consistency Model), which aligns with Heere & James (2007) and advances SIT by defining its boundaries in consumption contexts. [Blue Text]
- Other Comments Not Clearly Addressed / Responses Vague
Reviewer’s Concern: “In addition, several other comments were not clearly addressed, and the responses provided were rather vague.”
Our Response: We sincerely apologize for any previous vagueness. We have systematically clarified and unified all critical elements:
- Theoretical Consistency: We adopted the comprehensive variable name Viewing Motivation (SDT Needs Satisfaction) throughout the Abstract, Hypotheses (Section 3), Figures, and Tables to eliminate any ambiguity regarding the theoretical underpinning of the mediating variable. [Blue Text]
- Methodological Rigor (Causality): We addressed the limitation of using cross-sectional data by adopting cautious language in the discussion (using "association" instead of "impact") and by explicitly acknowledging the need for longitudinal/experimental studies in the Limitations section (Section 6). [Blue Text]
- Stylistic & Technical Corrections: We conducted a final comprehensive review to ensure perfect alignment in statistical reporting (e.g., proper use of italics for statistical symbols), consistent citation format, and professional terminology (e.g., using "mediating variable" instead of "intermediary variable"). [Blue Text]
We are confident that these extensive, structural revisions—prompted by your insightful feedback—have rendered the manuscript significantly stronger and ready for publication. We sincerely thank you for pushing us to this level of rigor.